# Rutaecarpine Ameliorates Murine N-Methyl-N’-Nitro-N-Nitrosoguanidine-Induced Chronic Atrophic Gastritis by Sonic Hedgehog Pathway

**DOI:** 10.3390/molecules28176294

**Published:** 2023-08-28

**Authors:** Yong He, Hong-Hong Liu, Xue-Lin Zhou, Ting-Ting He, Ao-Zhe Zhang, Xin Wang, Shi-Zhang Wei, Hao-Tian Li, Li-Sheng Chen, Lei Chang, Yan-Ling Zhao, Man-Yi Jing

**Affiliations:** 1School of Pharmacy, Chengdu University of Traditional Chinese Medicine, Chengdu 611137, China; heffly2000@hotmail.com (Y.H.); wangxin1214069407@163.com (X.W.);; 2Department of Pharmacy, Chinese PLA General Hospital, Beijing 100039, China; 3Department of Pharmacology, School of Basic Medical Sciences, Capital Medical University, Beijing 100039, China; 4School of Traditional Chinese Medicine, Southern Medical University, Guangzhou 510515, China

**Keywords:** rutaecarpine, chronic atrophic gastritis, flow cytometry, IL-1β, SHH signaling pathway, apoptosis

## Abstract

CAG is a burdensome and progressive disease. Numerous studies have shown the effectiveness of RUT in digestive system diseases. The therapeutic effects of RUT on MNNG-induced CAG and the potential mechanisms were probed. MNNG administration was employed to establish a CAG model. The HE and ELISA methods were applied to detect the treatment effects. WB, qRT-PCR, immunohistochemistry, TUNEL, and GES-1 cell flow cytometry approaches were employed to probe the mechanisms. The CAG model was successfully established. The ELISA and HE staining data showed that the RUT treatment effects on CAG rats were reflected by the amelioration of histological damage. The qRT-PCR and WB analyses indicated that the protective effect of RUT is related to the upregulation of the SHH pathway and downregulation of the downstream of apoptosis to improve gastric cellular survival. Our data suggest that RUT induces a gastroprotective effect by upregulating the SHH signaling pathway and stimulating anti-apoptosis downstream.

## 1. Introduction

Globally, CAG is a highly concerned and heavily burdensome disease, and the gastric mucosa is characterized by a decrease in the inherent glands [1]. As with intestinal metaplasia (IM), CAG is one of the primary precancerous lesions (GPL) in the pathogenesis from inflammation to gastric cancer (GC) [2,3]. In addition, the proportion of CAG in gastritis patients reached 25.8% in a multi-center study [4]. Therefore, it is essential to pay adequate attention to prevent the progression to GC.

An important cause of CAG is the excessive intake of N-nitrosamine in daily life [5]. MNNG is a kind of N-nitrosamine ingredient that has been used as a carcinogen in labs for 60 years and was also applied to imitate chemical nitrosamines in the daily diet [6]. DNA could obtain induced chromosomal abnormalities by MNNG. Therefore, it could also be used in gastrointestinal damage modeling and lead to the continuous development of dysplasia, intestinal metaplasia, CAG, GPL, and, finally, GC eventually [7]. Hence, the theoretical basis for successful CAG modeling by MNNG was supplied.

Proton-pump inhibitors (PPIs), H2 blockers, and anti-HP drugs have been commonly used, but the side effects and compliance issues due to long treatment cycles cannot be ignored. Additionally, numerous studies have reported a significant increase in serious adverse events and antibiotic resistance [8,9]. These were required for more medication protocols for CAG with around a 20% recurrence rate [10]. Finding adjuvant treatment methods and paying more attention to the effectiveness and safety issues [11] to alleviate the symptoms and improve the quality life of CAG patients were of great significance.

Some natural products with abundant sources and small side effects have been recognized by scientists during the increasing upgrade of molecular physicochemical research. Rutaecarpine (RUT), a quinazoline alkaloid, is isolated from the unripe Evodia Fructus. It has biological activities and pharmacological effects, including gastric mucosa protection, inflammatory inhibition, antioxidant, dilating uterus, brain function improvement, cardiovascular protection, etc. [12]. Numerous studies have confirmed that RUT demonstrated positive effects on HepG2 cells, macrophages, colitis, and gastric mucosa protection [13]. In addition, we have reported that RUT protects against ethanol-induced acute gastric ulcers before [14]. However, the chronic gastric-protective effect of RUT has been paid little attention.

In this study, we will focus on the therapeutic effect of the deleterious effects of MNNG on rat gastric tissue and attempt to elucidate the underlying mechanisms. RUT acts on MNNG-induced CAG model rats in vivo and GES-1 cellular damage in vitro (Figure 1) in our quest to discover a novel and candidate ingredient for further research on CAG treatment.

## 2. Results

### 2.1. RUT Mitigates MNNG-Co-Incubated GES-1 Cell Injury

GES-1 cell viability was measured using a CCK-8 kit. The optimum concentration of MNNG solution (10, 20, 30, 40, 50, 60, and 80 μM) on GES-1 cells at different points of time (6, 12, 24, 36, and 48 h) was explored. The best survival rate was defined as 60% [4,15]. Our results suggested that a MNNG solution of 40 μM and at a time of 24 h produced the most optical outcomes (Figure 2A). Furthermore, the protective concentration of RUT (2.0, 3.0, 4.0, 5.0, 8.0, and 10.0 μM) (Figure 2B) was explored. Finally, RUT at concentrations of 2.0, 3.0, 4.0, and 5.0 μM was selected for co-culture with 40 μM of MNNG for 24 h (Figure 2C). Our results indicated that the 4.0 μM RUT co-culture group presented a significant increase in survival rate compared with the MNNG group (*p* < 0.01). Therefore, 4.0 μM of RUT was selected as the dose for subsequent experiments in vitro.

### 2.2. RUT Alleviates MNNG Co-Cultured CAG by Activating Upregulation of SHH and Anti-Apoptosis Signaling

We further validated that RUT improved gastric mucosal injury induced by MNNG response by intervening with the SHH pathway and apoptosis-related mRNA expression, including SHH, Bcl-2, cyclin D1, Gli-1, Bax, IL-1β. Our results suggested that the intervention of RUT could increase the relative mRNA expression of SHH, Gli-1, and Bcl-2 significantly and may decrease the relative mRNA expression of Bax, cyclin D1, and IL-1β, perturbations of which were caused by MNNG co-cultured GES-1 cells (Figure 3).

### 2.3. RUT Protects against MNNG Co-Cultured Apoptosis in GES-1 Cells

GES-1 cells were pre-treated with 4.0 μM of RUT in the medium supplemented with 10% FBS for 2 h and then exposed to 40 μM of MNNG for an additional 24 h before being harvested for flow-cytometry analysis. Compared with the control cells, flow-cytometry analysis indicated that RUT intervention resulted in a decrease in MNNG-induced GES-1 cell apoptosis, as confirmed by the diminished fraction of GES-1 cells observed in the initial stages of the apoptotic process (Figure 4).

### 2.4. Effects of RUT on Macroscopic Pathology

After 12 weeks of MNNG administration combined with an irregulated diet, the gastric mucosa of the model group rats showed visible microinjury, thinning, paleness, and disarrayed plicae compared with the control group. After 4 weeks of RUT treatment via gavage, the rats were visibly well, and clinical signs, including diarrhea, weight loss, and loss of appetite, diminished. There was a significant change in body weight between the control group and the model group Additionally, based on two-way ANOVA analysis, we observed a statistically significant difference in body weight among the RUT-H, the RUT-L group, and the model group (*p* < 0.01). To explore the treatment effect of RUT on CAG rats, gastric tissues were collected, and HE staining was performed on the samples to detect the pathological changes. In the model group, gastric tissues, irregular arrangement of cells, inflammatory cell infiltration, and cystic dilation of tissue were found, in contrast to the control group, as illustrated in Figure 5. These changes improved after RUT treatment, as seen by a marked decrease in inflammatory cell infiltration and regular cellular arrangement in the tissue samples. These results indicated that the murine CAG model was successfully established, and RUT was observed to ameliorate histological changes in the gastric tissues of rats with CAG.

### 2.5. RUT Improves Gastric Mucosal Cells Apoptosis

To observe the apoptosis status of gastric mucosal cells, TUNEL staining was employed on the tissue samples, and the relevant results are shown in (Figure 6A). CAG-rat gastric mucosal cells in the model group exhibited a greater degree of apoptosis. After RUT treatment, the occurrence of apoptosis in gastric mucosal cells was reduced. Additionally, the expression of apoptosis-related proteins was identified by a western blotting assay. The levels of anti-apoptosis protein, Bcl-2, were found to be significantly decreased (*p* < 0.01). Meanwhile, proapoptotic proteins, including Bax and cleaved caspase-3, were observed to increase (*p* < 0.01) in model group rats that were administrated with MNNG combined with an irregular diet (Figure 6B). As expected, RUT treatment reversed the reduced Bcl-2 expression but enhanced cleaved-caspase-3 and Bax levels. The preceding observations suggest that RUT inhibits the apoptosis status of gastric mucosal cells in CAG rats.

### 2.6. RUT Affects the Serological Levels of Inflammatory Factors and Gastrointestinal Hormones

Serological levels of inflammatory factors and gastrointestinal hormones were measured in all rats. It was observed that in our murine models, the levels of PGI and PGI/PGII in serum were reduced, whereas IL-18, GAS-17, and IL-1β levels were enhanced after administration of MNNG combined with an irregular diet. After the administration of RUT, the levels of PGI and PGI/PGII were significantly increased (*p* < 0.05), and the levels of IL-18, GAS-17, and IL-1β decreased significantly (*p* < 0.05). The total data showed that RUT intervention could improve the inflammatory factors and gastrointestinal hormone levels in CAG rats (Figure 7).

### 2.7. Effect of RUT on SHH Pathway-Related Protein Expression

Initially, the model group rats displayed lower levels of SHH-related proteins, including SHH, SMO, and Gli-1, than those of the control group. After RUT administration, protein expression associated with the SHH pathway increased significantly (*p* < 0.01). The CAG rats displayed higher levels of cyclin D1 and IL-1β, compared with the control group. After RUT treatment, both these indicators were reduced in a significant manner (*p* < 0.05). This performance may verify activation of the SHH pathway by RUT treatment in CAG rats (Figure 8).

## 3. Materials and Methods

### 3.1. Drugs and Reagents

A component of RUT (CAS No. 84-26-4, purity ≥ 98%, No. CHB210904), ethyl carbamate (No. 20190419, chemically pure), was obtained from the Chemical Reagent of Sinopharm, and MNNG (CAS No. 70-25-7, purity ≥ 99.5%), was obtained from Chroma Biotechnology (Chengdu, China). MNNG and RUT were pretreated by dimethyl sulfoxide (DMSO) and diluted in required proportions in cell culture medium when co-incubated with GES-1 cells. Serological indicator kits were used for gastrin-17 (GAS-17, Cat. No. JM-10448R1), pepsinogen I (PGI, Cat. No. MM-0491R1), pepsinogen II (PGII, Cat. No. MG-0164R1), IL-1β (Cat. No. JM-01454R1), and IL-18 (Cat. No. MM-0194R1). All antibodies and their details are shown in Table 1. All other related products were obtained from commercial companies.

### 3.2. Ethics Approval Statement

All animal procedures were performed in line with the Guide for the Care and Use of Laboratory. This research was approved by the Ethics Committee of the Chinese PLA General Hospital (Ethics Approval No. IACUC-2022-010, Beijing, China).

### 3.3. Animal Handing

A total of 40 male Sprague-Dawley (SD) rats weighing between 200 and 220 g were purchased from the Beijing SiPeiFu Animal Breeding Center (Permission No. SCXK (Jing) 2019-0010, Beijing, China). The rats were fed adaptively for one week, then reared in specific pathogen-free (SPF) conditions, including humidity at 55 ± 5%, the temperature at 25 ± 0.5 °C, and a 12 h:12 h light–dark cycle. The rats were then randomly divided into five groups of eight animals each (Figure 9): control, CAG model, RUT 10 mg/kg/d (RUT-L), RUT 20 mg/kg/d (RUT-H) [16,17], and Vitacoenzyme 200 mg/kg/d [18]. The rats, except the control group, were free of water containing MNNG (170 μg/mL) [19,20] prepared in black bottles and were fed with an irregular diet for 12 weeks. Meanwhile, they were orally bred with water containing 4 °C MNNG (170 μg/mL) water every 48 h during the 12 weeks to build a CAG model. Subsequently, each experimental group was given their corresponding drugs over the next 4 weeks. The saline solution was fed to the control group, and the model rat group was provided with CMC-Na solution only [4]. After the final dose, animals were sacrificed. The gastric tissue and blood were collected. One hour after harvest, the blood was centrifuged at room temperature at 3000 rpm for 15 min. The supernatant was collected and stored at −80 °C before determining the serological parameters required by our investigation.

### 3.4. Determination of Pharmacodynamic Indicators

Serological indicators were identified via enzyme-linked immunosorbent assay (ELISA) kits (including GAS-17, PGI, PGII, IL-1β, and IL-18), which were supplied from Shanghai Meimian Biotech (Shanghai, China). All the operational processes and procedures were performed in strict accordance with the product instructions. After being placed in a 4% paraformaldehyde solution for 24 h, hematoxylin-and-eosin (HE) staining of gastric tissue was carried out for histological analysis. A microscope (Nikon Eclipse Ni-U) and Imaging Software of NIS-Elements 4.0 (Nikon, Japan) were used to observe and analyze the HE-stained sections and to photographically record analytical observations.

### 3.5. TUNEL Assay

TUNEL staining (Servicebio, Wuhan, China) of the gastric tissue was performed following the product specification. The gastric tissue sections were washed in PBS solution for three cycles. Subsequently, the samples were incubated at 37 °C for 0.5 h after 10% of 100 μL proteinase K solution was added and then washed with PBS three times. The samples were added to TdT enzyme solution (50 μL) and transferred to the tissue at 37 °C for an hour of incubation. After washing three times with PBS, the samples were mixed with 45 μL of labeling buffer and 5 μL of streptavidin fluorescein solution and stored in the dark at 37 °C for 0.5 h. The samples were then washed in PBS solution and stained in DAPI solution. The tissue sections were then observed and analyzed, and photographic images were recorded by a fluorescence microscope (Nikon, Japan).

### 3.6. Western Blotting Analysis

Western blotting analysis was employed as follows: gastric tissue proteins (40 mg) of the different groups were extracted and separated by sodium dodecyl sulfate-polyacrylamide gel (10%) electrophoresis (SDS-PAGE) buffer. According to the molecular weight (MW) of the target protein, the corresponding point dots of the gel were cut horizontally and transferred into the polyvinylidene difluoride (PVDF) membrane. Then, the protein membranes were incubated in a blocking buffer for 0.5 h, and the proteins were bound with their own specific antibodies, including SHH, Gli-1, SMO, Bcl-2, cyclin D1, caspase-3, IL-1β, cleaved caspase-3, and Bax at 4 °C overnight. Subsequently, the protein membranes were washed well, and the indicated secondary antibodies were added to and co-incubated with the membranes for 1.5 h at room temperature. Finally, a chemiluminescence kit (Tiangen, Beijing, China) and the gel image system were used to reveal the resultant protein bands. The ImageJ software of Version 1.53 was then utilized for analysis and quantification.

### 3.7. Conditions of GES-1 Cell Culture

GES-1 cells were purchased from the Fu Heng Cell Company (Shanghai, China). The cells were cultured in Dulbecco’s Modified Eagle’s Medium (DMEM, Gibco, Grand Island, NY, USA), together with 100 μg/mL of streptomycin, 100 units/mL of penicillin, and 10% fetal bovine serum (FBS). The dishes were incubated under the following conditions: 5% CO_2_ and 95% air humidified atmosphere at 37 °C in a cell culture chamber.

### 3.8. GES-1 Cell Viability Assay and Proliferation Analysis

GES-1 cell viability was monitored via an assay utilizing a CCK-8 kit. The cells (in petri dishes) were collected, centrifuged, and counted with a counting chamber. They were then diluted and distributed into 96-well plates with a cellular density of 5 × 10^3^ cells/well. After pretreating the cells with RUT solution for 2 h, MNNG was added, and the cells were co-incubated for 24 h. CCK-8 test solution was then added to every well, and the wells were co-incubated for 1h. Eventually, a Synergy TM H1 instrument (BioTek, Shoreline, WA, USA) was used to detect absorbance at 450 nm. The optical density (OD) value was used for calculating the viability percentage of GES-1 cells. Each protocol was repeatedly operated three times under identical conditions.

### 3.9. Flow-Cytometry Analysis

Flow-cytometry (BD FAC Symphony A5, Franklin Lakes, NJ, USA) analysis was employed to ascertain the apoptosis status of GES-1 cells. The cells were distributed into 6-well plates at a density of 2 × 105/well. While at an 80% confluency of cells, they were pretreated with the RUT solution for 2 h. Subsequently, MNNG solution was added to and co-incubated with cells in an incubator at 37 °C in 5% CO_2_ for 24 h. Finally, the cells were collected and then stained by the Annexin V/7AAD Kit in accordance with the reagent instructions.

### 3.10. Extraction of RNA and Analysis by qRT-PCR

The TRIzol reagent (Invitrogen, Carlsbad, CA, USA) was used for total RNA extraction. A Primer Script RT Reagent Kit was employed to reverse transcribe RNA into synthesized cDNA. Relative RNA expression levels were quantified by quantitative real-time PCR using the SYBR Green Super Mix Kit (Servicebio, Wuhan, China) on an AB7300 thermocycler (Biosystems, Boston, MA, USA), combined with customized primers (Table 2). GAPDH served as the internal reference, and the 2^−ΔΔCt^ protocol was utilized to calculate the relative expression levels of mRNA.

## 4. Discussion

Cancer caused by chemical compounds in the diet and in the environment is a complex and multistage process whereby normal cells, over an extended period of exposure, eventually transform into cancer cells. This induced process can be caused by a driver of harmful chemicals, such as nitrosamines, polycyclic aromatic hydrocarbons, azo dyes, etc. [21] The normal regulation of gastric cellular differentiation, proliferation, apoptosis, and the self-stability of gastric tissue is disrupted by mutational genetic changes induced by longstanding internal and external factors. The preceding knowledge provides an evidence base for the utilization of MNNG administration in rats, combined with an irregular diet, to establish a murine CAG model.

Two important pro-inflammatory factors, IL-1β and IL-18, are involved in immune responses and cellular anti-inflammatory activities, including cellular immunity, proliferation, pyroptosis, and apoptosis. Both IL-18 and IL-1β are members of the IL-1 superfamily, which are closely associated with acute and chronic inflammation and are mainly involved in Th2-type immune responses and the inflammatory response [22]. As such, they are often used in research to evaluate the severity of gastric mucosal injury [23]. In the present study, ELISA tests were utilized to observe an upregulation of IL-18 and IL-1β in the model group and also a significant decrease (*p* < 0.01) in these levels after RUT administration. GAS-17 is secreted by G cells of the gastrointestinal system and when combined with PGI and PGI/PGII, are likely to reflect the relative degree of atrophy of the gastric antrum and gastric body [24,25]. The changes of these factors in the present study indicate an improvement in CAG after RUT treatment. The expression of cyclin D1 should not be detected in normal gastric mucosa, and there are significant trends (*p* < 0.01) for increased expression of cyclin D1 in non-neoplastic mucosa, including in conditions such as gastric atrophy, dysplasia, intestinal metaplasia, and gastritis to gastric cancer [26]. Our study indicates that the distinct levels of expression of cyclin D1 reflect different stages of CAG severity and also illustrates the therapeutic effect of RUT on rats with CAG. In the present study, high-dose RUT had the most noticeable mitigating effect on the degree of atrophy of gastric mucosal cells.

The SV40T gene was integrated into the GES-1 cell genome and expressed in the nucleus, which was based on human gastric mucosa and could be cultured in vitro. The in vitro gastric-mucosal cell damage model was established via MNNG treatment of GES-1 cells, and consequently, the morphology and growth characteristics of the model cells changed, compared with normal GES-1 cells. The proliferation, differentiation, and epithelial regeneration of GES-1 model cells are known to be regulated by the SHH signaling pathway [23,27]. The severity of gastric atrophy may be influenced by the inhibition of the SHH signaling pathway by IL-1β, which may inhibit gastric-acid secretion and intracellular calcium release (Figure 10). The corpus glands that are closer to the rat forestomach exhibit consistently high levels of expression of SHH protein [13]. Additionally, IL-1β is the strongest inhibitor of gastric acid found thus far, and the continuous secretion of low levels of gastric acid into the stomach will promote the occurrence and development of atrophy and lead to an increase in the risk of development of GC [28]. In the present study, SHH-associated proteins or mRNAs, including SHH, SMO, and Gli-1, have been detected to have a decreased tendency among model rats and MNNG-co-incubated GES-1 cells. The proteins and mRNA expression of SHH, Gli, and SMO are upregulated after RUT administration, and their returns indicate a reduction in mucosal cell damage. These results concur with those of Shiotani [29,30].

Apoptosis in gastric mucosal cells is the basis of carcinogenesis, and therefore, apoptosis inactivation plays a critical role in aborting the progression of CAG to GC [31] Notably, the inhibition of SMO is eliminated, which leads to an upregulation of Gli-1 during the appearance of the SHH-signaling peptide. Subsequently, more Gli-1 factors emerge and enter the nucleus, and apoptosis-associated genes and proteins are downregulated [32,33]. Caspase-3 is located downstream of the caspase cascade reaction and is regarded as a true enforcer for the induction of apoptosis [32], while cleaved caspase-3 is the active expression of caspase-3. Therefore, cleaved-caspase-3 detection is often adopted as a precise indicator for the determination of apoptosis [33]. We observed that cleaved-caspase-3 protein expression was increased, while the expression of total caspase-3 was unchanged. We thus attempted to understand the effects of RUT on gastric mucosa apoptosis by determining apoptosis-associated proteins and the mRNA of Bcl-2, cleaved caspase-3, and Bax. It was observed that levels of cleaved caspase-3 and Bax significantly (*p* < 0.01) increased, and Bcl-2 levels decreased in the model group compared with the control group. These results suggest that the rats that were administered MNNG combined with an irregular diet activated cleaved-caspase-3 expression and apoptosis in vivo. This concurs with the findings of studies conducted by Liu and Ren [16,34] The present study, thus, attempts to elucidate the therapeutic effects and the underlying mechanisms of action of RUT in the treatment of a murine model of CAG. We believe that our work has created a credible and valid foundation for the future investigation of novel therapeutics based on RUT for the effective treatment of CAG.

## 5. Conclusions

In summary, the current results showed that RUT indeed significantly improved CAG-rat gastric damage. These effects are associated with upregulating the Sonic Hedgehog pathway and downstream of apoptosis. We envisage that the data above could be provided for follow-up studies on RUT.

## Figures and Tables

**Figure 1 molecules-28-06294-f001:**
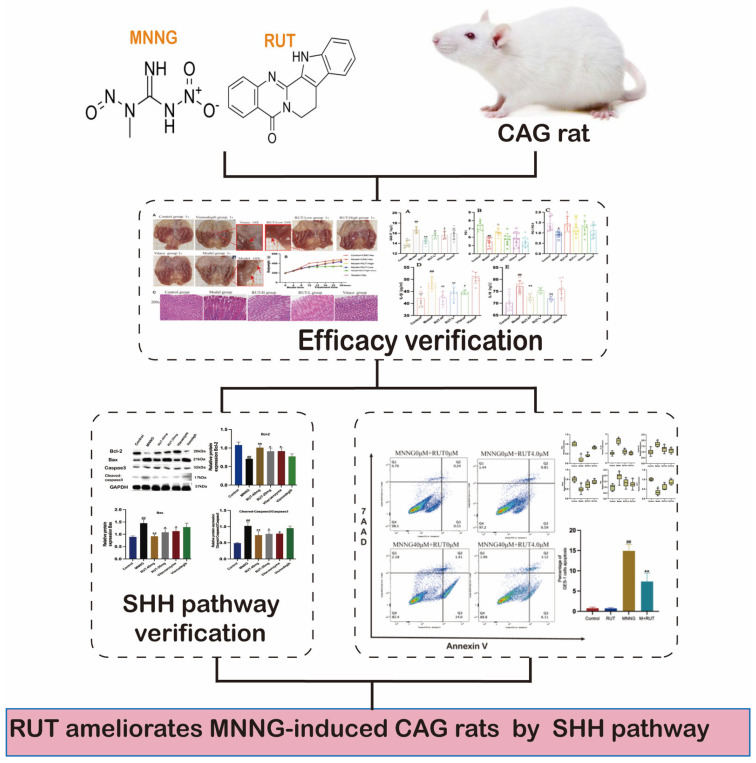
Flow chart of this study.

**Figure 2 molecules-28-06294-f002:**
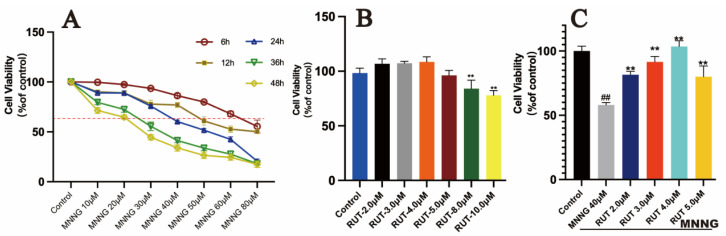
Effect of MNNG and RUT on cell viability of GES-1 cells. (**A**) Cell survival rate of GES-1 cells treated with different doses of MNNG for different times. (**B**) Cell survival rate of GES-1 cells treated with different doses of RUT. (**C**) Protective effect of RUT pretreatment on the proliferation of MNNG co-cultured GES-1 cells. Data are shown as mean ± SD (*n* = 6). ** *p* < 0.01 vs. model; ^##^
*p* < 0.01 vs. control.

**Figure 3 molecules-28-06294-f003:**
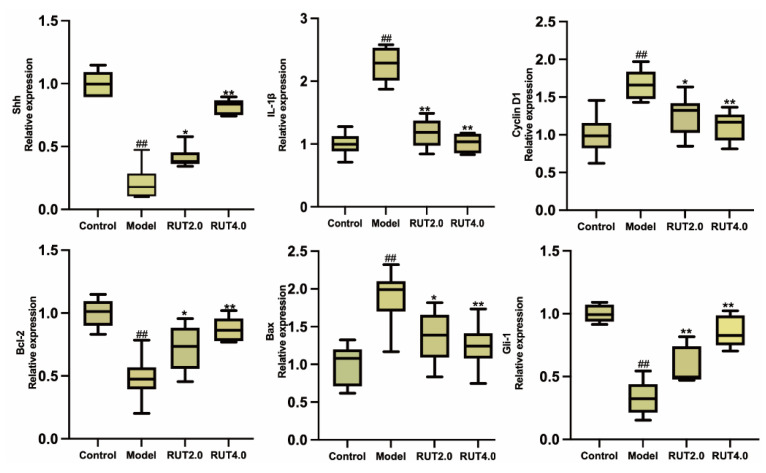
Histogram of SHH, Gli-1, IL-1β, Bcl-2, Bax, and cyclin D1 mRNA expression in MNNG co-cultured GES-1 cells by qRT-PCR. * *p* < 0.05, ** *p* < 0.01 vs. model; ^##^
*p* < 0.01 vs. control (*n* = 6).

**Figure 4 molecules-28-06294-f004:**
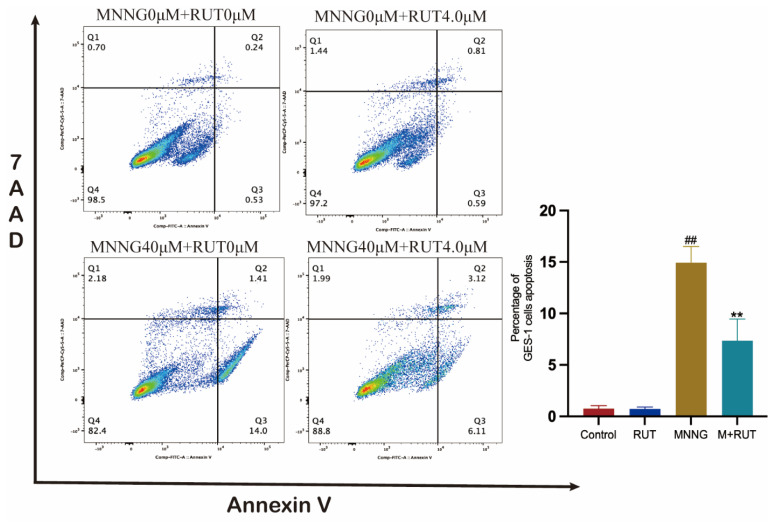
RUT protects against MNNG-induced apoptosis in GES-1 cells by flow-cytometry analysis (*n* = 6). Data are shown as mean ± SD (*n* = 6). ** *p* < 0.01 vs. model; ^##^
*p* < 0.01 vs. control.

**Figure 5 molecules-28-06294-f005:**
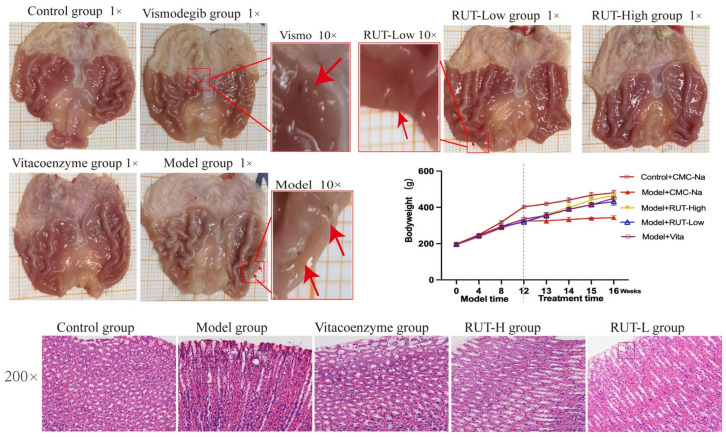
Effects of RUT on macroscopic pathological changes of gastric-mucosa gastric tissue in CAG rats (red arrow marks micro injury). Bodyweight of rats during the modeling and treatment period. Data are expressed as mean ± SD (*n* = 8). RUT-relieved histological lesions of gastric tissues, with representative images of HE staining from each experimental group (magnification ×200).

**Figure 6 molecules-28-06294-f006:**
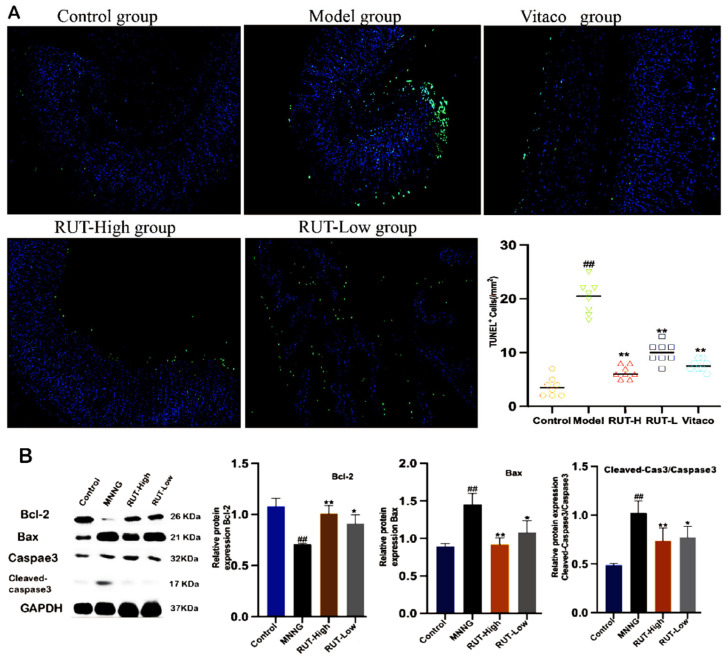
RUT-inhibited apoptosis of gastric mucosal cells (**A**). RUT affected the expression of apoptosis-associated proteins in gastric tissues (**B**). The expression levels of Bcl-2, Bax, and cleaved caspase-3 were measured using western blotting. * *p* < 0.05, ** *p* < 0.01 vs. model; ^##^
*p* < 0.01 vs. control (*n* = 3).

**Figure 7 molecules-28-06294-f007:**
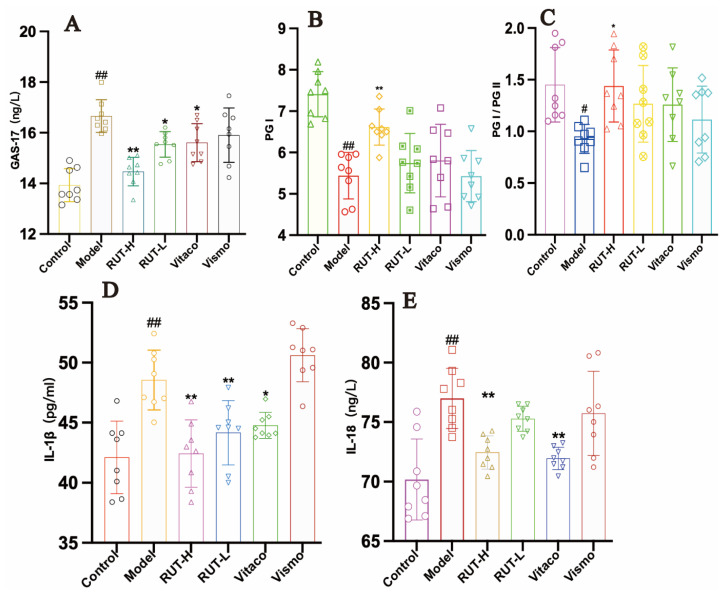
RUT regulated the serological levels of gastrointestinal hormones and inflammatory cytokines in CAG rats induced by MNNG combined with an irregular diet, including (**A**) GAS-17, (**B**) PGI, (**C**) PG1/PGII, (**D**) IL-β, and (**E**) IL-18 detected using ELISA kits. * *p* < 0.05, ** *p* < 0.01 vs. model; ^##^
*p* < 0.01 vs. control (*n* = 8).

**Figure 8 molecules-28-06294-f008:**
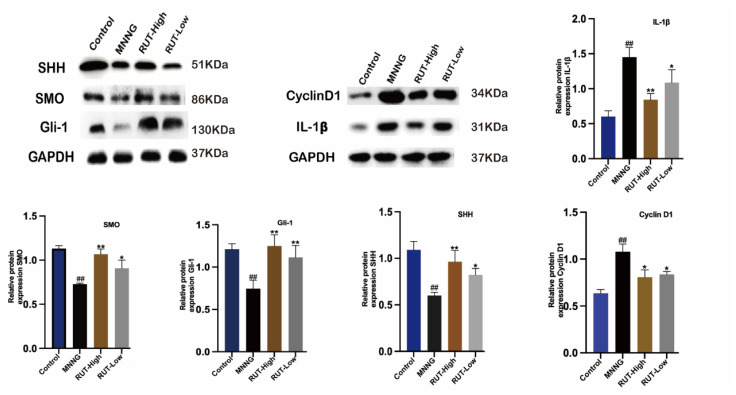
RUT affected the SHH pathway-related protein expression in CAG rats induced by MNNG combined with an irregular diet. * *p* < 0.05, ** *p* < 0.01 vs. model; ^##^
*p* < 0.01 vs. control (*n* = 3).

**Figure 9 molecules-28-06294-f009:**
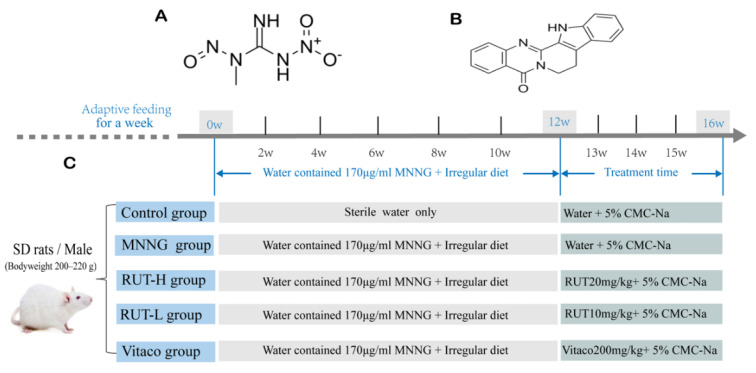
Experimental design flow chart of the induction of MNNG and drug treatment. The chemical structure of MNNG (**A**). The chemical structure of RUT (**B**). Experimental design flow chart of the induction of MNNG and drug treatment (**C**).

**Figure 10 molecules-28-06294-f010:**
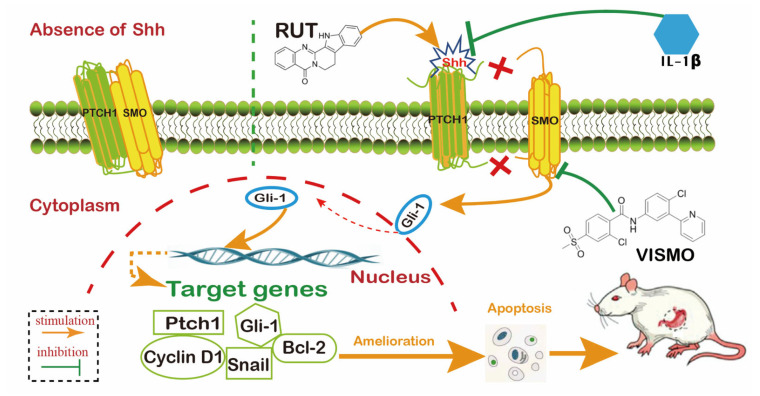
Schematic diagram of RUT amelioration of MNNG-induced chronic gastric mucosal injury by upregulating IL-1β-inhibited Sonic Hedgehog pathway.

**Table 1 molecules-28-06294-t001:** Antibody information.

Antibody	Dilution	Manufacturer	Cat. No.
Rabbit anti-SHH	1:1000	BOSTER BIOTECH	D-B1-07G07A
Rabbit anti-IL-1β	1:1000	BOSTER BIOTECH	BOS703BP70
Rabbit anti-Bax	1:1000	BOSTER BIOTECH	BA0315-2
Rabbit anti-Bcl-2	1:1000	BOSTER BIOTECH	A00040-2
Rabbit anti-Gli-1	1:1000	Bioss Antibodies	BS-1206R
Rabbit anti-SMO	1:1000	Beytime Biotech	AF-7836
Rabbit anti-cyclin D1	1:1000	BOSTER BIOTECH	BST17044272
Rabbit anti-caspase-3	1:1000	BOSTER BIOTECH	BM3957
Rabbit anti-cleaved caspase-3	1:5000	Abcam	[E83-77] ab32042
GAPDH monoclonal antibody	1:10,000	Proteintech	60004-1-lg
Goat anti-rabbit IgG (H + L)	1:10,000	ZSGB-BIO	ZB-2301

**Table 2 molecules-28-06294-t002:** Primer sequences of qRT-PCR analyses for mRNA expression.

Genes	Forward	Reverse
GAPDH	5′-GGAAGCTTGTCATCAATGGAAATC-3′	5′-TGATGACCCTTTTGGCTCCC-3′
Bcl-2	5′-GGAGGATTGTGGCCTTCTTTG-3′	5′-AGACAGCCAGGAGAAATCAAACA-3′
Bax	5′-CGGGTTGTCGCCCTTTTCTA-3′	5′-GAGGAAGTCCAATGTCCAGCC-3′
Gli-1	5′-CTGACGCCCATGTGACCAA-3′	5′-TGCAAGGTCCCTCGTCCAA-3′
SHH	5′-AGGAGTGAAACTGCGGGTGA-3′	5′-CACCGAGCAGTGGATATGTGC-3′
Cyclin D1	5′-AGCTGTGCATCTACACCGAC-3′	5′-GAAATCGTGCGGGGTCATTG-3′
IL-1β	5′-CGATCACTGAACTGCACGCTC-3′	5′-ACAAAGGACATGGAGAACACCACTT-3′

## Data Availability

The data underlying this article are available in the article.

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
