# Peer review of "Rutaecarpine Ameliorates Murine N-Methyl-N’-Nitro-N-Nitrosoguanidine-Induced Chronic Atrophic Gastritis by Sonic Hedgehog Pathway"

_molecules, 2023, doi:10.3390/molecules28176294_

Round 1
Reviewer 1 Report
He et. al showed a that Rutaecarpine can improve CAG, similar to what they have previously shown where RUT exerted a gastroprotective effect against gastric mucosal injury induced by ethanol. The underlying mechanism might be associated with the improvement of anti-inflammatory and anti-apoptosis mechanism. The current manuscript shows for the first time that RUT somehow improves CAG but restoring the SHH signalling and activating anti-apoptosis mechanism. However, the histological pictures presented were not so convincing. Is there a dose dependent effect of RUT in vivo? It was also not clear whether RUT+ SHH antagonist showed the same phenotype as the model. It would be good if the authors quantify the histological score of each rat.
It will also help to strengthen the conclusion of the manuscript by adding Vismodegib +MNNG control group in the in vitro experiments.
Patched1 inhibition is an important readout to indicate the activation of HH signalling. It would help if the authors also include this.
The language use is fine but requires minor editing.
Author Response
Dear editor,
It is a great honor for us to revise the manuscript entitled “Rutaecarpine Ameliorates Murine MNNG-induced Chronic Atrophic Gastritis by Sonic Hedgehog Pathway
” (Manuscript NO: molecules-2423288). We have carefully revised the manuscript and resubmitted it to Molecules online in with your kind suggestions. The response letter had contained a point-by-point reply to the peer comments. Please see the revised manuscript. Thank you so much for your time and your work on this manuscript. The main corrections in the paper and response to the reviewer’s comments are as follows:
Comments from the reviewers:
-Reviewer 1
We are truly grateful for your time and effort in thoroughly reviewing and evaluating the manuscript, as your constructive feedback has been instrumental in shaping the final edition of this paper. Our response to your question is as follows:
*1. the histological pictures presented were not so convincing
Reply: We took this recommendation seriously and reconstructed the pathology correlates. The new image shows the damage of gastric tissue more clearly, and I hope it will be recognized by the reviewers.
*2 Is there a dose dependent effect of RUT in vivo?
Reply:The doses of RUT20mg/kg/d and 10mg/kg/d in vivo experiments were observed only, so we did not investigate the dose-dependent issue. If there will be a follow-up study, we will pay more attention to the dose-dependent design.
*3 Patched1 inhibition is an important readout to indicate the activation of HH signalling. It would help if the authors also include this.
Reply: Indeed, we were originally designed to detect the WB validation of the patch1 protein. The positive rate was not good enough due to the molecular weight of this protein was 124-128 kDa and lack of the quality of the specific antibody. In order to ensure the objectivity and authenticity of the results, we abandoned it. However, the lack of patch1 protein does not affect our analysis of the results of SHH pathway due to the other important proteins results are positive.
Reviewer 2 Report
The manuscript needs minor revision. Please see comments given in the text of reviewed attached file of the manuscript.

Author Response
Dear editor,
It is a great honor for us to revise the manuscript entitled “Rutaecarpine Ameliorates Murine MNNG-induced Chronic Atrophic Gastritis by Sonic Hedgehog Pathway
” (Manuscript NO: molecules-2423288). We have carefully revised the manuscript and resubmitted it to Molecules online in with your kind suggestions. The response letter had contained a point-by-point reply to the peer comments. Please see the revised manuscript. Thank you so much for your time and your work on this manuscript. The main corrections in the paper and response to the reviewer’s comments are as follows:
Comments from the reviewers:
-Reviewer 2
We are truly grateful for your time and effort in thoroughly reviewing and evaluating the manuscript, as your constructive feedback has been instrumental in shaping the final edition of this paper. Our response to your question is as follows:
*1. The manuscript needs minor revision. Please see comments given in the text of reviewed attached file of the manuscript.
Reply: Thanks very much for the opinions and suggestions of the reviewers, we will re-polish the manuscript to avoid any typographical and grammatical errors.
Reviewer 3 Report
The research title “Rutaecarpine Ameliorates Murine MNNG-induced Chronic 2 Atrophic Gastritis by Sonic Hedgehog Pathway”
1. Appropriately titled and well written the whole manuscript
2. N-methyl-N'-nitro-N-nitrosoguanidine (MNNG) administration combined with an irregular diet to successfully establish a CAG rat model.
3. RUT treatment showed positive effects on CAG rats by improving the histological damage of gastric tissue and causing changes in serological indicators.
4. In in vitro experiments, RUT increased the proliferation of GES-1 cells and ameliorated the damage induced by MNNG.
5. The protective effect of RUT on gastric mucosal damage and GES-1 cells was associated with the upregulation of the Sonic Hedgehog (SHH) signaling pathway.
6. RUT treatment resulted in the upregulation of anti-apoptotic proteins (Bcl-2) and downregulation of pro-apoptotic proteins (Bax and Caspase 3), indicating its potential in improving gastric cellular survival.
7. The study provides robust and reproducible laboratory-based evidence supporting RUT as a potential treatment for chronic atrophic gastritis.
After Line number 81, need to cite the following reference:
1. Subramaniyan V, Fuloria S, Gupta G, Kumar DH, Sekar M, Sathasivam KV, Sudhakar K, Alharbi KS, Al-Malki WH, Afzal O, Kazmi I, Al-Abbasi FA, Altamimi ASA, Fuloria NK. A review on epidermal growth factor receptor's role in breast and non-small cell lung cancer. Chem Biol Interact. 2022; 351:109735.
After Line number 386, need to cite the following reference:
2. Fuloria S, Subramaniyan V, Karupiah S, Kumari U, Sathasivam K, Meenakshi DU, Wu YS, Guad RM, Udupa K, Fuloria NK. A Comprehensive Review on Source, Types, Effects, Nanotechnology, Detection, and Therapeutic Management of Reactive Carbonyl Species Associated with Various Chronic Diseases. Antioxidants (Basel). 2020;9(11):1075.
These findings suggest that Rutaecarpine (RUT) has therapeutic potential for the treatment of chronic atrophic gastritis by improving gastric tissue damage, promoting cell survival, and modulating key signaling pathways.
Report:
After revising the suggested comments, this can be published in the journal of molecules.

Author Response
Dear editor,
It is a great honor for us to revise the manuscript entitled “Rutaecarpine Ameliorates Murine MNNG-induced Chronic Atrophic Gastritis by Sonic Hedgehog Pathway
” (Manuscript NO: molecules-2423288). We have carefully revised the manuscript and resubmitted it to Molecules online in with your kind suggestions. The response letter had contained a point-by-point reply to the peer comments. Please see the revised manuscript. Thank you so much for your time and your work on this manuscript. The main corrections in the paper and response to the reviewer’s comments are as follows:
Comments from the reviewers:
-Reviewer 2
We are truly grateful for your time and effort in thoroughly reviewing and evaluating the manuscript, as your constructive feedback has been instrumental in shaping the final edition of this paper. Our response to your question is as follows:
*1. After Line number 81, need to cite the following reference:
- Subramaniyan V, Fuloria S, Gupta G, Kumar DH, Sekar M, Sathasivam KV, Sudhakar K, Alharbi KS, Al-Malki WH, Afzal O, Kazmi I, Al-Abbasi FA, Altamimi ASA, Fuloria NK. A review on epidermal growth factor receptor's role in breast and non-small cell lung cancer. Chem Biol Interact. 2022; 351:109735.
After Line number 386, need to cite the following reference:
- Fuloria S, Subramaniyan V, Karupiah S, Kumari U, Sathasivam K, Meenakshi DU, Wu YS, Guad RM, Udupa K, Fuloria NK. A Comprehensive Review on Source, Types, Effects, Nanotechnology, Detection, and Therapeutic Management of Reactive Carbonyl Species Associated with Various Chronic Diseases. Antioxidants (Basel). 2020;9(11):1075.
These findings suggest that Rutaecarpine (RUT) has therapeutic potential for the treatment of chronic atrophic gastritis by improving gastric tissue damage, promoting cell survival, and modulating key signaling pathways.
Reply: Thank you very much for the comments and suggestions of the reviewers. We have added the above two references to the corresponding positions.
Round 2
Reviewer 2 Report
The manuscript can be accepted for publication.